# Antifeeding and Oviposition Deterrent Effect of *Ludwigia* spp. (Onagraceae) against *Plutella xylostella* (Lepidoptera: Plutellidae)

**DOI:** 10.3390/plants11192656

**Published:** 2022-10-10

**Authors:** Eliana Aparecida Ferreira, Eduardo Carvalho Faca, Silvana Aparecida de Souza, Claudemir Antonio Garcia Fioratti, Juliana Rosa Carrijo Mauad, Claudia Andrea Lima Cardoso, Munir Mauad, Rosilda Mara Mussury

**Affiliations:** 1Laboratory of Insect-Plant Interaction, Graduate Program in Entomology and Biodiversity Conservation, College of Biological and Environmental Sciences, Federal University of Grande Dourados, Dourados-Itahum Highway, 12th km, Dourados 79804-970, Mato Grosso do Sul, Brazil; 2Graduate Program in Agribusiness, Federal University of Grande Dourados, Dourados-Itahum Highway, 12th km, Dourados 79804-970, Mato Grosso do Sul, Brazil; 3Laboratory of Chemistry, State University of Mato Grosso do Sul, Dourados-Itahum Highway, 12th km, Dourados 79804-970, Mato Grosso Do Sul, Brazil; 4Laboratory of Vegetables Production, Faculty of Agricultural Science, Federal University of Grande Dourados, Dourados-Itahum Highway, 12th km, Dourados 79804-970, Mato Grosso do Sul, Brazil

**Keywords:** antixenosis, plant extract, diamondback moth, phagodeterrent, botanical derivaties

## Abstract

Plants produce a wide variety of bioactive compounds with insecticidal properties, such as secondary metabolites capable of interfering with the nutrition and reproduction of pest species such as *Plutella xylostella*. Thus, the objective of this study was to evaluate the effects of aqueous and ethanolic extracts of *Ludwigia* spp. (Onagraceae) on the feeding and oviposition of *P. xylostella.* Choice bioassays were performed using aqueous and ethanolic extracts. The aqueous extract of *L. tomentosa* resulted in an approximately 81% reduction in larval feeding compared to that in the control, with an antifeedant index (AI) of 52%. The aqueous and ethanolic extract of *L. nervosa* acted by stimulating larval feeding. The oviposition was significantly reduced in the kale leaves treated with aqueous and ethanolic extracts of *Ludwigia* spp. The aqueous extracts promoted an average 90% reduction in oviposition when compared to that in the control, and an oviposition deterrent index (ODI) above 61% was classified as an oviposition deterrent. In addition, ethanolic extracts affected 81% of oviposition, with an ODI above 41%. Bioassays should be performed to clarify the use of aqueous and ethanolic extracts of *L. nervosa* as they acted as phagostimulants in the feeding tests and as deterrents in the oviposition tests. The phenolic compounds—flavonoids, condensed tannins, and alkaloids—were more abundant in *L. nervosa, L. tomentosa, L. sericea*, and *L. longifolia*. The extracts of *L. longifolia* and *L. tomentosa* showed the best results, interfering with the host choice for feeding and oviposition in *P. xylostella* and representing an alternative for the control of diamondback moths.

## 1. Introduction

The diamondback moth (*Plutella xylostella* L.; Lepidoptera: Plutellidae) is globally recognized as an important pest species due to the substantial damage it causes to the culture of brassica, represented by vegetables such as cabbage, cauliflower, broccoli, and kale, among others [1,2]. This damage usually results in the depreciation of the product in the market and interferes with the growth of the plant, causing death or total loss of production [3]. The status of this pest is maintained in the absence of effective natural enemies and its ability to develop resistance to synthetic pesticides [4], which are the main tactics for pest control [5].

The rapid life cycle of *P. xylostella*, associated with the constant selection pressure caused by repeated applications of pesticides, has allowed the insect to evolve resistance to a wide range of pesticides—including *Bacillus thuringiensis* toxins [6]. This resistance, together with consumer demand for products with low chemical residue, highlights the need for new compounds to control this insect [4].

As a result, there has been an increase in the search for alternative methods of pest control, including biopesticide control. To date, there are four categories of effective commercial botanical products: pyrethrins, rotenones, azadirachtins, and essential oils [7]. The control of insects using compounds derived from plants represents an important alternative in integrated pest management because plants produce a wide variety of bioactive compounds with insecticidal properties, such as secondary metabolites capable of interfering with nutrition, development, reproduction and survival [8,9,10], deterrents, toxicity, sterility, and growth [11,12,13]. Although many of these substances of natural origin are subject to rapid environmental degradation, they contain numerous chemicals with different and complex mechanisms of action that delay the evolution of resistance to insecticides [14]. These bioactive substances are perceived by insects through the sensilla, specialized structures of the epidermis that act as receptors that perceive a variety of environmental stimuli, mediating the behavior of insects [15].

*Ludwigia* spp. can be found in much Brazilian territory, in wet or flooded areas [16]. Species of this genus are reported in the literature as having medicinal [17], antibacterial [18], antioxidant [19], antifungal [20], and insecticidal properties [10]. Phytochemical screenings have shown that species *Ludwigia adscendens* (L.) H. Hara contains flavonoids such as quercetin, in addition to terpenes, triterpenoids, other phenols, tannins, and alkaloids [21]. *Ludwigia octavalvis* (Jacq.) P. H. Raven has flavonoids, phenols, saponins, steroids, and tannins as secondary metabolites [18]. However, *Ludwigia abyssinica* A.Rich and *Ludwigia decurrens* Walter have revealed the presence of alkaloids and tannins only [20]. Phytochemical screenings have shown that the species *Ludwigia tomentosa* (Cambess.) H. Hara, *Ludwigia longifolia* (DC.) H. Hara, *Ludwigia sericea* (Cambess.) H. Hara, and *Ludwigia nervosa* (Poir.) H. Hara have secondary metabolites such as flavonoids, phenolic compounds, condensed tannins, and alkaloids, and that these substances that are able to inhibit food consumption and interfere with the morphological and physiological transformations of *P. xylostella* offspring and oviposition with exposure to aqueous extracts during the larval stage [10].

Considering that understanding the extracts’ action mode allows them to be used more efficiently and that there is no information on *Ludwigia’s* effect on host choices for feeding and oviposition, we propose to test the following hypothesis: aqueous and ethanolic extracts of *Ludwigia* spp. cause a larvae antifeeding effect and the deterring of oviposition in *P. xylostella* females.

## 2. Results

### 2.1. Feeding Preference

In the choice test for the aqueous extract, we observed that *L. nervosa* and *L. longifolia* presented negative AI values, stimulating feeding of the larvae; in contrast, the aqueous extract of *L. tomentosa* (W = 85.00; *p* = 0.0088) presented an AI of 52.13%, standing out among the other treatments as a phagodeterrent. In addition, in comparison to the control, *L. tomentosa* reduced the leaf area consumed by 80.92% (Table 1).

There was also no significant difference in the leaf area consumption among the ethanolic extract treatments and the control. The ethanolic extract of *L. nervosa*, as in the bioassay with aqueous extract, acted by stimulating the feeding of *P. xylostella* larvae (Table 1). The most significant AI value was found for *L. sericea* (29.42%), followed by that of *L. longifolia* (23.79%); both were classified as phagodeterrents (Table 1).

Comparing the types of solvent, we observed that only *L. longifolia* showed variation in its effect, and in the choice test, that the aqueous extract acted as a phagostimulant; at the same time, the ethanolic extract reduced feeding of the larvae, acting as phagodeterrent (Table 1).

### 2.2. Oviposition

In the choice test, the number of eggs oviposited by the *P. xylostella* females was lower for all extracts when compared to that with the control treatment (χ^2^ = 10.317, *df* = 4, *p* = 0.0354). The aqueous extracts of the *Ludwigia* spp. reduced oviposition in *P.* xylostella females by 89.41% on average. However, *L. longifolia* and *L. tomentosa* were the treatments that obtained the lowest mean number of eggs, with a reduction in oviposition of 97.16% and 91.29%, respectively (Table 2). In comparison to the control, *L. nervosa* and *L. sericea* promoted a reduction in oviposition greater than 80% (Table 2).

Additionally, the aqueous extracts—especially *L. longifolia* and *L. tomentosa—*were responsible for ODI values above 60%, with ODI values of 76.42% and 70.69%, respectively (Table 2). In general, the aqueous extracts were classified as oviposition deterrents because they were ODI positive.

In the choice test, we observed that females had oviposition preferences for discs treated with distilled water (control). There was a significant difference between the treatments with ethanolic extract and the control treatment (χ^2^ = 11.880, *df* = 4, *p* = 0.0182), and the ethanolic extracts of the *Ludwigia* spp. reduced oviposition by 81.24% on average (Table 2).

In comparison to the control, the *L. longifolia* and *L. nervosa* treatments most affected oviposition, reducing oviposition by 91.19% and 81.29%, respectively; consequently, these treatments had ODI values of 73.70% and 60.02%, respectively (Table 2). In general, the ethanolic extracts obtained an ODI above 40%, indicating a deterrent effect of the extracts and, consequently, a preference for oviposition by discs treated with distilled water (Table 2).

Antioxidant activity was more prominent in *L. longifolia*, *L. sericea*, *L. tomentosa*, and *L. nervosa*, in that order. Phenolic compounds, flavonoids, condensed tannins, and alkaloids were found in greater quantities in the extracts of *L. nervosa*, *L. tomentosa*, *L. sericea*, and *L. longifolia*, in that order (Table 3).

## 3. Discussion

Phytochemical screening showed that *Ludwigia* spp. presented all classes of compounds investigated—our results can be attributed to some of these classes. *Ludwigia* spp. contain substances that make it difficult to eat, prolonging the larval stage due to a lower ingested food conversion [22]. This can occasionally lead to the death or reduced growth of larvae [23], pupal survival [24], impaired feeding, digestion inhibition, and the release of free radicals [25]. The compounds most reported in the literature that affect insect feeding or oviposition belong to the alkaloid, flavonoid, terpenoid, and phenol groups [26,27]. Flavonoids such as quercetin 3-arabinoside, quercetin 3-glucoside, and quercetin 3-rutinoside have already been identified in some species of *Ludwigia* [28] and have been found to be able to act as phagodeterrents, depending on the concentrations used [29]. However, other types of flavonoids can cause adult mortality and reduce oviposition and larval emergence when eggs come into contact with these substances [26]. Thus, the secondary compounds present in the studied species of the genus *Ludwigia* compromised both the feeding and oviposition of *P. xylostella.*

Our observations are supported by the papers of Ferreira et al. [10] and Ferreira [30], where the authors analyzed theses extracts’ actions in relation to the insect life cycle. Looking at the bioassays performed by Ferreira et al. [10] and the results obtained in the present study leads us to the interpretation that phagostimulant actions—especially in aqueous extracts of *L. longifolia—*compromises the development of individuals, with a significant reduction in pupal weight.

In the Ferreira [30] studies, it was observed that ethanol extracts resulted in a reduction in pupal weight—especially for *L. tomentosa—*which may be due to the phagodeterrence of the extract, as observed in this study from the low consumption of treated discs—showing that the ethanolic extracts of *Ludwigia* spp. were, for the most part, less preferred for consumption. The reduction in leaf consumption when insects are in contact with plant extracts is usually the result of phagodeterrents or sublethal intoxication, preventing feeding and digestion [7]. In our results, similarly to Ferreira [30], we found that the extract of *L. tomentosa* presented better results in the bioassays of food preference, being responsible for reductions in the consumption of leaves—which in the field would characterize a good form of control since the leaves are a commercial product in cabbage cultures.

For aqueous and ethanolic extracts of *L. nervosa*, a food stimulus was observed, causing the larvae of the *P. xylostella* to consume more treated discs than control discs. This result can be attributed to the sensitivity of the larvae to different allelochemicals and the larvae-perceived kairomones—substances that stimulate feeding—causing the larvae to bite the discs test and continue feeding [26]. Some types of flavonoids, such as apigenin and naringenin [29], can induce variable behavioral responses and can also act as phagostimulants or phagodeterrents [31]. The high consumption of some extracts—when they act as phagostimulants—is not always advantageous for insects, since the increased intake of some phytochemicals can result in sublethal effects that impair their development.

In relation to oviposition tests with aqueous and ethanolic extracts, females of *P. xylostella* oviposited preferentially in the control discs, indicating a deterrent effect on oviposition. The aqueous and ethanolic extracts of *L. longifolia* caused the highest deterrence rates in the bioassays. When an insect lands on a plant, the perception of its chemical and physical characteristics is of great importance for the occurrence or lack of oviposition [32]. Thus, several factors may have contributed to the deterrence of oviposition: The first factor resulted from the presence of phytochemicals in the extracts of *Ludwigia* spp. that caused alterations in the physiology and behavior of *P. xylostella* adults, reflected in the oviposition response. The other factor was the presence of impeding substances in the extract, restricting the arrival of moths to the oviposition substrate [33]. The deterrence of oviposition was observed by other authors when evaluating the effects of botanical extracts of *Melia azedarach, Azadirachta indica* [34], and *Pachyrhizus erosus* against *P. xylostella* [35].

In the use of ethanolic extracts of *Ludwigia* species on the life cycle of *P. xylostella*, Ferreira [30] observed that the mean number of newly emerged larvae was significantly reduced for *L. sericea*. In this study, when considering egg survival, there was no significant difference—although it was reduced in all treatments with ethanolic extracts. For the aqueous extract, the ovideterrence observed in the present work is reflected in the research of Ferreira et al. [10], who observed that egg survival was reduced by *L. sericea* extracts and that in the adult phase of *P. xylostella—*in addition to a reduction in fecundity and the number of newly emerged larvae for all treatments—there was a significant difference for the extract of *L. tomentosa*.

Considering previous studies involving phytochemical analyses with aqueous extracts of *Ludwigia* spp. [10], when compared with the results found for the ethanolic extract, there is a pattern in relation to the quantification of secondary compounds for each species. The different responses obtained from the bioassays are due to the specificity of extraction of secondary compounds that each solvent has, which results in different responses regarding foliar consumption and oviposition in *P. xylostella*.

## 4. Materials and Methods

The extracts were prepared and the bioassays were conducted in the Laboratory of Insect–Plant Interactions of the School of Biological and Environmental Sciences at the Federal University of Grande Dourados (Universidade Federal de Grande Dourados-UFGD) in Dourados, Mato Grosso do Sul, Brazil. The bioassays and rearing occurred in a controlled environment with a constant temperature of 25 ± 2 °C, a relative humidity of 55 ± 5%, and a photoperiod of 12h. The cabbage leaves used in the experiments were of approximately the same age—that is, around 60 days after planting, and were acquired in local gardens.

### 4.1. Collection of Botanical Material

Fully expanded leaves of *Ludwigia tomentosa* (Cambess.) H. Hara, *Ludwigia longifolia* (DC.) H. Hara, *Ludwigia sericea* (Cambess.) H. Hara, and *Ludwigia nervosa* (Poir.) H. Hara were collected in a transition area between the Atlantic Forest and cerrado forest in Dourados, Mato Grosso do Sul (22°11′54.92″ S; 54°46′52.15″ W). The botanical material was collected in the morning in the autumn seasons of the years 2018 and 2019, during the phenological stages of flowering and fruiting. The plants were identified by a specialist and deposited in the herbarium at the UFGD with the following numbers: 6391-*L. tomentosa,* 6389-*L. longifolia*, 6388-*L. sericea*, and 6390-*L. nervosa*. The collection of botanical material was authorized by the National Council for Scientific and Technological Development (Conselho Nacional de Desenvolvimento Científico e Tecnológico-CNPq) and the Council for the Management of Genetic Heritage (CGEN/MMA) under the number A9ECAC6.

### 4.2. Preparation of Botanical Material

The leaves collected previously were sanitized and dried in a forced air oven for 72 h at 40 °C ± 1 °C. Subsequently, the leaves were ground in an industrial mill, and 5 g of plant matter was added to 50 mL of distilled water. The solution remained in a refrigerated environment at 10 °C for 24 h and was then filtered with the aid of filter paper to obtain crude aqueous extracts at a concentration of 10% (*w/v*).

### 4.3. Preparation of Ethanolic Extracts

The sanitized leaves were dried in a forced air oven for three days at a maximum temperature of 40 °C (±1 °C). After this period, the leaves were ground in an industrial mill. The resulting powder was added to ethanol (95%) at a ratio of 75 g of plant to 300 mL of solvent. Every seven days, the solution was filtered and an additional 300 mL of ethanol was added. The aforementioned process was performed 5 consecutive times, totaling 1.5 L of extract. The filtered ethanolic extract was concentrated in a rotary evaporator at 60 °C under reduced pressure. Subsequently, the concentrate was solubilized in distilled water at a concentration of 0.6%.

### 4.4. Breeding of Plutella xylostella

The breeding of *P. xylostella* (Figure 1) was performed from pupae collected in an organic planting area of *Brassica oleracea* var. *acephala,* located in the municipality of Dourados, Mato Grosso do Sul, Brazil, following the methodology adapted from Barros et al. [36].

### 4.5. Bioassay of Antifeedant Activity in P. xylostella

For the with choice bioassays, 3rd instar *P. xylostella* larvae fasted for 4 h. The kale discs (*B. oleracea* var. *Acephala*, 4 cm in diameter) were immersed in the respective treatments (aqueous and ethanolic extracts of 4 plants of the genus *Ludwigia*) and control (distilled water) for one minute and subsequently distributed in plastic trays to dry naturally for 20 min.

In the choice test, four kale discs were distributed in the Petri dish (9 cm in diameter and 1.5 cm in height) in a crosswise and equidistant manner—two of which were immersed in the respective extracts, and the other two immersed in distilled water (control treatment). The kale discs were placed under a moistened filter paper disc (9 cm in diameter) (Figure 2). Subsequently, three *P. xylostella* third instar larvae were added to the Petri dish, remaining there for 48 h. The bioassays occurred in a controlled environment with a constant temperature of 25 ± 2 °C, a relative humidity of 55 ± 5%, and a photoperiod of 12 h.

To evaluate the consumption of *P. xylostella* larvae in both tests, the leaf discs were scanned to verify the leaf area consumed using ImageJ software [38] and the antifeedant index (AI) [33,39,40] was calculated.

### 4.6. Plutella xylostella Oviposition Deterrent Bioassay

Multiple-choice oviposition bioassays were performed with *P. xylostella* adults up to 12 h old, and they were kept in the laboratory without contact with the extracts. Leaf discs of kale were immersed in the different treatments, and after drying naturally for 20 min, were placed in plastic cages.

In the multiple-choice test, three pairs of *P. xylostella* were transferred to a plastic cage (30 cm long × 15 cm wide × 12 cm high) containing six treated kale discs (4 cm in diameter) arranged equidistantly in a circular shape (Figure 3), with one leaf disc from each extract treatment (four species of *Ludwigia*) and two discs from the control treatment (distilled water).

The pairs of *P. xylostella* were kept for four days in oviposition cages and were fed a 10% diluted honey solution soaked in cotton and changed daily. The eggs were counted with the aid of a stereoscopic microscope at 24, 48, 72, and 96 h, and at each interval, the leaf discs were replaced by newly treated discs and the oviposition deterrent index (ODI) of Huang and Renwick [41] was calculated. The bioassays were carried out in a controlled environment with relative humidity of 55 ± 5%, a constant temperature of 25 ± 2 °C, and a photoperiod of 12 h.

### 4.7. Phytochemical Analysis of the Extract

The phytochemical screenings for the aqueous extract are described in Ferreira et al. [10] and the analyses of the ethanol extract were performed on the same plant samples over the same collection period.

#### 4.7.1. Phenolic Compounds, Flavonoids, and Tannins

The samples were prepared at a concentration of 1000 µg/mL for analysis. The tests were evaluted in triplicate.

The phenolic compounds were determined employing the Folin–Ciocalteu reagent method and the absorbance at 760 nm was measured using a spectrophotometer (FEMTO 700 PLUS, FEMTO, São Paulo, São Paulo, Brazil) [42]. Gallic acid (Sigma-Aldrich, St. Louis, MO, USA) was used as a standard at concentrations of 5–1000 µg/mL. The results are expressed in milligrams of gallic acid per gram of dry weight of extract.

The flavonoids were determined using the AlCl_3_ reagent method. The absorbance at 430 nm was measured using a spectrophotometer (FEMTO 700 PLUS) [41]. Rutin (Sigma-Aldrich, St. Louis, MO, USA) was used as a standard at concentrations of 1–50 µg/mL. The results are expressed in milligrams per gram of dry weight of extract.

The condensed tannin was determined using the vanillin method [42]; the absorbance was measured at 510 nm [43]. Rutin (Sigma-Aldrich, St. Louis, MO, USA) was used as a standard at concentrations of 0.1–50 µg/mL. The results are expressed in catechin milligrams per gram of dry weight of extract.

#### 4.7.2. Determination of Alkaloid Content

The total alkaloid content in the samples was quantified according to the procedure developed by Oliveira et al. [44] and the absorbance at 435 nm was measured. Berberine (Sigma-Aldrich, St. Louis, MO, USA) was employed as the standard and linearity was obtained between 40 and 200 µg/mL. The results are expressed in berberine milligrams per gram of dry weight of extract.

#### 4.7.3. Determination of Antioxidant Activity

The antioxidant activity of the extracts was assessed using the free radical indicator DPPH (1,1-diphenyl-2-picrilhidrazyl; Sigma-Aldrich, St. Louis, MO, USA) [45]. The extracts were prepared with distilled water at the following concentrations: 5, 10, 20, 200, 30, 40, 50, 60, 70, 80, 90, and 100 µg/mL. The percentage of inhibition by each concentration was used to obtain the IC_50_ values.

### 4.8. Statistical Analysis

#### 4.8.1. Feeding Bioassay

The experimental design was completely randomized, consisting of five treatments (4 extracts and 1 control) and 10 replicates for each type of extract (aqueous and ethanol). All data were subjected to the Shapiro–Wilk normality test. For the choice bioassay, the Mann–Whitney U test was used to compare the control and treatment groups. For all tests, the significance level was set at 5%.

#### 4.8.2. Antifeedant Index (AI)

The AI was calculated using the formula AI = (C − T)/(C + T) *100, where C represents the control and T is the treated leaf area consumed by the larvae [26,27,46]. This index distinguishes whether an extract is a phagostimulant (negative values) or phagodeterrent (positive values). The treatments that were statistically insignificant were still labeled as a phagodeterrent and/or phagostimulant.

#### 4.8.3. Oviposition Bioassay

The oviposition bioassay was performed in a completely randomized experimental design consisting of five treatments, four extracts, and one control (distilled water). The treatments consisted of 10 replicates, i.e., each replicate consisted of 1 cage. All data were subjected to the Shapiro–Wilk normality test. As the data from the multiple-choice bioassays did not show a normal distribution, the data were subjected to analyses of variance, and the means were compared by the Kruskal–Wallis test. For all tests, the significance level was set at 5%.

#### 4.8.4. Oviposition Deterrent Index (ODI)

The ODI was calculated according to Huang and Renwick [41] as follows: ODI = (Cn − Tn)/ (Cn + Tn)*100, where Cn represents the number of eggs deposited in the control and Tn represents the number of eggs oviposited in the treated leaves. If the ODI is greater than zero, then it is classified as a deterrent; if the ODI is equal to zero, then it is neutral; and if the ODI is negative, then it is considered a stimulant. The treatments that are statistically insignificant are still labeled as a deterrent and/or stimulant.

#### 4.8.5. Chemical Analysis

The results were submitted to the Kruskal–Wallis test (*p* ≤ 0.05). All phytochemical screenings were performed in triplicate and the results are expressed as mean ± confidence interval (95%). Data were analyzed on the R platform and p values less than 0.05 (*p* < 0.05) were considered indicative of significant differences between the samples compared in each test.

## 5. Conclusions

Leaf consumption was reduced when the *L. tomentosa* extract was used. *L. longifolia* and *L. nervosa* stimulated larval feeding. The ethanolic extracts were classified as phagodeterrents—with the exception of *L. nervosa*, which was a phagostimulant. All extracts were classified as oviposition deterrents in *P. xylostella* females, regardless of the solvent used. There was a significant reduction in egg oviposition for both types of extracts, especially *L. tomentosa* and *L. longifolia*. Besides this, phytochemical screening showed that the ethanolic *Ludwigia* extracts contained phenolic compounds, flavonoids, condensed tannins, and alkaloids—substances that are able to interfere with feeding and oviposition in *P. xylostella*, representing an alternative for the control of diamondback moths.

## Figures and Tables

**Figure 1 plants-11-02656-f001:**
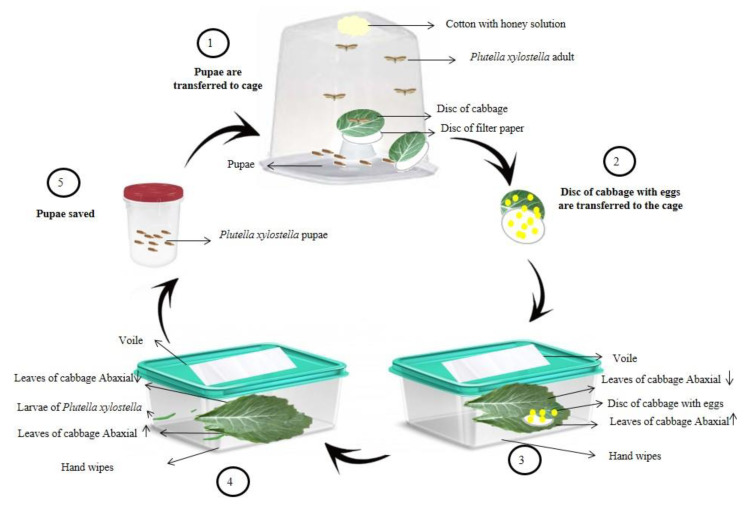
Schematic representation of the methodology adapted for the breeding of *Plutella xylostella.* Source: Image adapted from Matias da Silva et al. [37].

**Figure 2 plants-11-02656-f002:**
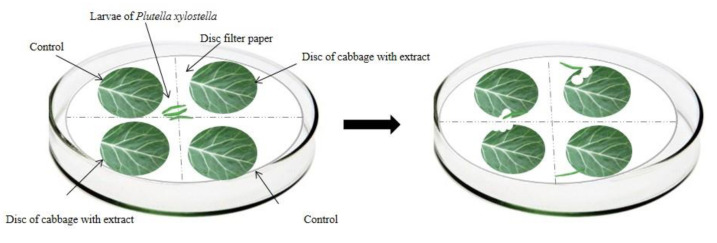
Schematic representation of the choice bioassay of antifeedant activity for *Plutella xylostella* larvae.

**Figure 3 plants-11-02656-f003:**
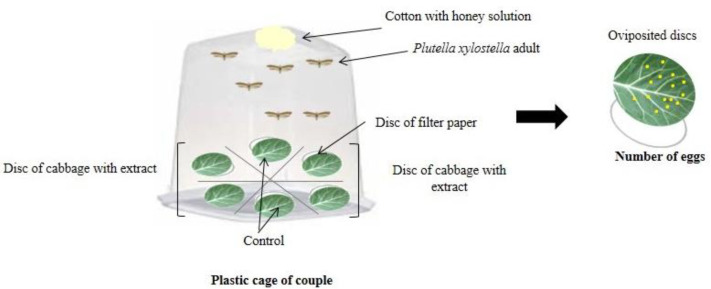
Schematic representation of the multiple-choice oviposition bioassays.

**Table 1 plants-11-02656-t001:** Leaf area consumed by *Plutella xylostella* larvae and antifeedant index (AI) values resulting from aqueous extracts of *Ludwigia* spp. after food exposure.

Treatment	Leaf Area Consumed (cm^2^)		*p*-Value	AI %	Classification
**AQUEOUS EXTRACT**					
**Choice Bioassay**					
	**Control**	**Extract**			
*L. tomentosa*	0.42 ± 0.12 A	0.08 ± 0.03 B	0.0088	52.13	Phagodeterrent
*L. longifolia*	0.15 ± 0.07 A	0.09 ± 0.02 A	0.8451	(−)15.53	Phagostimulant
*L. sericea*	0.21 ± 0.06 A	0.10 ± 0.03 A	0.3254	11.49	Phagodeterrent
*L. nervosa*	0.11 ± 0.03 A	0.23 ± 0.06 A	0.1387	(−) 27.58	Phagostimulant
**ETHANOLIC EXTRACT**					
**Choice** **Bioassay**					
	**Control**	**Extract**			
*L. tomentosa*	0.31 ± 0.10 A	0.23 ± 0.06 A	0.7617	8.18	Phagodeterrent
*L. longifolia*	0.28 ± 0.05 A	0.22 ± 0.07 A	0.3442	23.79	Phagodeterrent
*L. sericea*	0.52 ± 0.09 A	0.32 ± 0.11 A	0.1506	29.42	Phagodeterrent
*L. nervosa*	0.42 ± 0.13 A	0.36 ± 0.09 A	0.9705	(−) 2.16	Phagostimulant

Means followed by different letters in the column differ at the 5% significance level. SE—standard error; AI—antifeedant index.

**Table 2 plants-11-02656-t002:** Mean number of eggs oviposited by *Plutella xylostella* in the choice bioassays and the oviposition deterrent index (ODI) based on the use of aqueous and ethanolic extracts of *Ludwigia* spp.

Treatment	Number of Eggs ± SE	*p*-Value	ODI %	Classification
**AQUEOUS EXTRACT**				
**Choice** **Bioassay**				
Control	49.40 ± 17.07 a	0.0335	- -	
*L. tomentosa*	4.30 ± 2.02 b		70.69	Deterrent
*L. longifolia*	1.40 ± 0.65 b		76.42	Deterrent
*L. sericea*	9.20 ± 3.74 b		61.23	Deterrent
*L. nervosa*	6.00 ± 2.85 b		66.88	Deterrent
**ETHANOLIC EXTRACT**				
**Choice Bioassay**				
Control	64.70 ± 14.56 a	0.0182	- -	
*L. tomentosa*	14.90 ± 6.33 b		60.02	Deterrent
*L. longifolia*	5.70 ± 1.96 b		73.70	Deterrent
*L. sericea*	15.8 ± 5.80 b		49.64	Deterrent
*L. nervosa*	12.1 ± 4.90 b		41.73	Deterrent

Means followed by different letters in the column differ at a 5% significance level. SE—standard error; ODI—oviposition deterrent index.

**Table 3 plants-11-02656-t003:** Antioxidant activity data (IC_50_—minimum inhibitory concentration), phenolic compounds, flavonoids, condensed tannins, and alkaloids of ethanolic extracts of *Ludwigia* spp.

Extract	Antioxidant Activity	Phenolic Compounds	Flavonoids	Condensed Tannins	Alkaloids
	IC50 (μg mL^−1^)	(mg g^−1^)	(mg g^−1^)	(mg g^−1^)	(mg g^−1^)
*L. tomentosa*	44.7 ±0.4	189.8 ±2.8	123.7 ±1.3	33.8 ±0.3	15.9 ±0.2
*L. longifolia*	49.7 ± 0.2	182.4 ±1.1	101.1 ± 1.1	30.1 ± 0.4	14.8 ±0.2
*L. sericea*	47.8 ± 0.5	179.6 ± 1.3	117.6 ± 0.9	32.3± 0.7	15.3 ± 0.3
*L. nervosa*	41.4 ± 0.5	201.1 ± 2.1	132.9 ± 1.2	34.9 ± 0.6	16.7 ± 0.5

## Data Availability

The data presented in this study are available on request from the corresponding author.

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
