# Peer review of "Antifeeding and Oviposition Deterrent Effect of Ludwigia spp. (Onagraceae) against Plutella xylostella (Lepidoptera: Plutellidae)"

_plants, 2022, doi:10.3390/plants11192656_

Round 1
Reviewer 1 Report
In the submitted manuscript, the authors aimed to evaluate the effect of aqueous and ethanolic extracts of Ludwigia spp. (Onagraceae) on the feeding and oviposition of P. xylostella using No choice and choice bioassays. As a result, the extracts of the Ludwigia spp. were found to interfere with the choice of host for feeding and oviposition by P. xylostella, and may be representing an alternative in the control of diamondback moth. However, the work is not well presented, and some issues needed to be addressed. Thus, major revisions are suggested.
Some detailed comments are made as below:
1). The phytochemicals in the aqueous and ethanolic extracts of Ludwigia spp. should be analyzed using LC-MS or other analytical methods, which may better explain the effects of these extracts.
2). In the Results part, more experiments should be done and added, for example effects of different partition fractions should be compared, together with their phytochemicals. In the Discussion part, some deeper discussion about the potential active compounds in the corresponding extracts or fractions should be added to this part.
3). In the conclusion part, only the tests on the aqueous and ethanolic extracts are obviously not enough for an article, the corresponding phytochemicals and the underlying mechanisms are a must for this study.
4). Many English writing errors are found, and should be corrected. Please check the writing format of the whole manuscript carefully.
Reviewer 2 Report
This manuscript describes the phenomenology of feeding and oviposition responses by Plutella xylostella when exposed to aqueous and ethanolic extracts of Ludwigia spp. on host kale disks.
The manuscript is well-written and logical.
My major comment that you might consider is to include a graphical depiction of your results that are presented in tabular form. Graphical data would more clearly convey the effects of your treatments on feeding and oviposition.
Another comment is that you might include a sentence or two explaining why treatments that are statistically insignificant are still labeled as phagodeterrent/deterrent/phagostimulant.
Editorial comments:
line 26: "(FI)" should be changed to "(AI)"
line 43: delete "others" after period
line 137: delete last comma and "among"
line 138: delete "the species"
Reviewer 3 Report
The authors of this manuscript attempt to evaluate the effect of aqueous and ethanolic extracts of Ludwigia spp. (Onagraceae) on the feeding and oviposition of P. xylostella.
I think it could be an interesting article, and represents a scientific contribution related to the scope of the Plants journal. Nevertheless, there are critical questions and corrections before the manuscript should be accepted for publications. I therefore recommend its publication with deep major corrections.
The issues that need to be addressed are listed below:
1. If the ODI and AI parameters are used for activity assignment, statistical processing of their values is desirable.
2. "The extracts of Ludwigia spp. interfere with the choice of host for feeding and oviposition of P. xylostella, representing an alternative in the control of the diamondback moth". This statement is not supported by the results obtained, especially, if the residual activity of the treatments has not been studied.
3. The authors should include in more detail previous studies on the phytochemistry of the different Ludwigia spp. Fundamentally because the efficacy of the different extracts is highly conditioned by the species.
4. Ethanolic and aqueous extracts of have to be tested on non targets insects including parasitoids before being included in integrated pest management programs currently being developed. Please exclude their inclusion in IPM programs, without mentioning the need for this study.
5. Why did they choose a 0.6% solution? Would it not have been more convenient to use different doses, so that the results would be more consistent? Are your conclusions only valid with that concentration?
Is it necessary to include experiments with different doses of extracts
Reviewer 4 Report
Accepted with modification:.
In the Introduction, it is necessary to mention the secondary metabolites of the plants of Ludwigia (Ludwigia tomentosa; Ludwigia longifolia; Ludwigia longifolia y Ludwigia nervosa), because in this job phytochemical screening of Ludwigia spp.
Methodology
4.1. At what time of the year did they collect botanical material? At what phenological stage of the plants?
4.4. The characteristics of the plant are not mentioned. How many days were they growing? Are the environmental characteristics of the place missing, the coordinates?
Section 4.5 is repeated:
4.5. Bioassay of Antifeedant Activity in P. xylostella
4.5. P. xylostella Oviposition Deterrent Bioassay, Chance 4.6
4.6. Statistical Analysis, Chance 4.7
Environmental conditions missing in two sections: Bioassay of Antifeedant Activity in P. xylostella and P. xylostella Oviposition Deterrent Bioassay.
Replicates of non-choice bioassays should be duplicated.
Results
It strikes me that the results of no choice do not show significant differences, it would be appropriate to double the replicates.
Discussion
It should be clear that you did not phytochemical screening of Ludwigia spp., nor did they conduct electrophysiological studies on the antennae of the adults, which are the ones that search for a host. In addition, the antennal structures of the larvae are very different from those of the adults.
In lines 172-175, where you talk about the larvastatic effect, you did not study this effect, which would be important to address.
In the bioassays you cut the cabbage leaves, you caused mechanical damage and in your discussion you do not mention this effect. I recommend you read:
Reyes-Prado, H., Jiménez-Pérez, A., Arzuffi, R., and Robledo, N. 2020. Copitarsia decolora Guenée (Lepidoptera: Noctuidae) females avoid larvae competition by detecting larvae damaged plants. Scientific Reports, 10(1), 1-7. ISSN 2045-2322. DOI: 10.1038/s41598-020-62365-5
Round 2
Reviewer 1 Report
The authors have tried their best to address most of the issues raised, and the revised version is improved, and thus acceptable in its current form.
Author Response
We appreciate the reviewer valuable suggestions.
Reviewer 3 Report
I thank the authors for modifying the results and conclusions in the new version of the manuscript.
Author Response

(The authors gave the same response as above.)
